# Fluctuations in Upper and Lower Body Movement during Walking in Normal Pressure Hydrocephalus and Parkinson’s Disease Assessed by Motion Capture with a Smartphone Application, TDPT-GT

**DOI:** 10.3390/s23229263

**Published:** 2023-11-18

**Authors:** Chifumi Iseki, Shou Suzuki, Tadanori Fukami, Shigeki Yamada, Tatsuya Hayasaka, Toshiyuki Kondo, Masayuki Hoshi, Shigeo Ueda, Yoshiyuki Kobayashi, Masatsune Ishikawa, Shigenori Kanno, Kyoko Suzuki, Yukihiko Aoyagi, Yasuyuki Ohta

**Affiliations:** 1Department of Behavioral Neurology and Cognitive Neuroscience, Tohoku University Graduate School of Medicine, Sendai 980-8575, Japan; s-kanno@med.tohoku.ac.jp (S.K.); kyon@med.tohoku.ac.jp (K.S.); 2Division of Neurology and Clinical Neuroscience, Department of Internal Medicine III, Yamagata University School of Medicine, Yamagata 990-2331, Japan; toshikon@med.id.yamagata-u.ac.jp (T.K.); yasuyuki@med.id.yamagata-u.ac.jp (Y.O.); 3Department of Informatics, Faculty of Engineering, Yamagata University, Yonezawa 992-8510, Japan; t222164m@st.yamagata-u.ac.jp (S.S.); fukami@yz.yamagata-u.ac.jp (T.F.); 4Department of Neurosurgery, Nagoya City University Graduate School of Medical Sciences, Nagoya 467-8601, Japan; shigekiyamada393@gmail.com; 5Interfaculty Initiative in Information Studies, Institute of Industrial Science, The University of Tokyo, Tokyo 113-8654, Japan; 6Normal Pressure Hydrocephalus Center, Rakuwakai Otowa Hospital, Kyoto 607-8062, Japan; rakuwadr1001@rakuwadr.com; 7Department of Anesthesiology, Yamagata University School of Medicine, Yamagata 990-2331, Japan; hayasakatatsuya1101@gmail.com; 8Department of Physical Therapy, Fukushima Medical University School of Health Sciences, 10-6 Sakaemachi, Fukushima 960-8516, Japan; mhoshi@fmu.ac.jp; 9Shin-Aikai Spine Center, Katano Hospital, Katano 576-0043, Japan; uedashigeo@yahoo.co.jp; 10Human Augmentation Research Center, National Institute of Advanced Industrial Science and Technology (AIST), Kashiwa II Campus, University of Tokyo, Kashiwa 277-0882, Japan; kobayashi-yoshiyuki@aist.go.jp; 11Rakuwa Villa Ilios, Rakuwakai Healthcare System, Kyoto 607-8062, Japan; 12Digital Standard Co., Ltd., Osaka 536-0013, Japan; y.aoyagi@digital-standard.com

**Keywords:** fractality, 1/f noise, smartphone device, gait analysis, neurodegenerative diseases

## Abstract

We aimed to capture the fluctuations in the dynamics of body positions and find the characteristics of them in patients with idiopathic normal pressure hydrocephalus (iNPH) and Parkinson’s disease (PD). With the motion-capture application (TDPT-GT) generating 30 Hz coordinates at 27 points on the body, walking in a circle 1 m in diameter was recorded for 23 of iNPH, 23 of PD, and 92 controls. For 128 frames of calculated distances from the navel to the other points, after the Fourier transforms, the slopes (the representatives of fractality) were obtained from the graph plotting the power spectral density against the frequency in log–log coordinates. Differences in the average slopes were tested by one-way ANOVA and multiple comparisons between every two groups. A decrease in the absolute slope value indicates a departure from the 1/f noise characteristic observed in healthy variations. Significant differences in the patient groups and controls were found in all body positions, where patients always showed smaller absolute values. Our system could measure the whole body’s movement and temporal variations during walking. The impaired fluctuations of body movement in the upper and lower body may contribute to gait and balance disorders in patients.

## 1. Introduction

Describing and measuring the gaits of patients have long been clinical challenges. In previous studies, terms describing pathological gait, such as short stepped, shuffling, freezing, wide based, festination, hemiplegic, spastic, ataxic, and instability, have been useful and identifiable characteristics for diagnosing diseases, such as Parkinson’s disease (PD) [1,2,3,4], cerebrospinal degeneration [4,5,6], and idiopathic normal pressure hydrocephalus (iNPH) [7,8,9,10]. These characteristics are observed in aspects such as the length, speed, timing, and laterality of movement in the feet and legs within the range of human visual perception. Clinicians learn from textbooks and examine patients for many years; however, we are often unable to reach an agreement on descriptions of gait [8]. The quantitative dynamic evaluation of gait has also been difficult because of the uncertainty of human vision. Therefore, studies have tried to assess the cycle and variability of gait with various sensors, such as floor sheets [5,6,11], wearable devices [12,13], and motion-capture systems [14,15].

To introduce our method for measuring gait dynamics, let us mention the fluctuations found in nature. Nature hosts some intriguing examples of self-similar structures, such as the Roman cauliflower, in which almost exact copies of the entire flower may be recognized on multiple smaller scale examples, and the correlations between the frequency and size of the copy structure express the 1/f pattern [16]. We can also observe the same pattern in many biological phenomena and complex systems in nature [17,18,19]. Human motion is diverse and contains inherent variability, yet it is not entirely random. For instance, when standing still in a relaxed state, the swaying motion exhibits fractal, 1/f properties [20], exhibiting a roughly proportional relationship between log power and log frequency in a spectral power analysis. In other words, 1/f relations refer to self-similar fluctuations observed across multiple time scales, indicating fractal patterns. By applying fractality, healthy and pathological gaits, as well as their differences, have been explored in the field of bioengineering [21,22,23,24,25,26,27,28,29]. These studies assessed the dynamic aspect of gait, with a focus on inconsistent fluctuation, which was different from classical quantitative analysis centered on calculating averages of key parameters, such as stride length, swing, and speed, to establish stable and consistent patterns in gait. Hausdorff et al. proposed watching the stride-to-stride fluctuations while considering that the fluctuations are not just noise but convey important information [21,30]. In healthy adults, the fluctuations from stride to stride are not completely random; instead, each stride time is related to adjacent stride times and to stride times hundreds of strides later [22]. The distribution of strides in a healthy walk has a 1/f-like structure [31]. This pattern of fluctuation (noise) has also been referred to as pink noise. In contrast, noise that is unhealthy, unstable, and lacks a fractal pattern is referred to as white noise.

The advancements in these sensors and methods for capturing the dynamics of human body movements have significantly contributed to the current wealth of knowledge. As a result, our goal is to make these beneficial sensors as easily accessible as possible for patients in need. The system we employed was a smartphone-based markerless motion-capture system using artificial intelligence (AI), named TDPT-GT (Three-Dimensional Pose Tracker for Gait Test), which essentially consisted of taking videos of the patients on an iPhone [32,33,34]. A noninvasive motion-capture application allowed us to use this system in the general room for outpatients. The patient setup required only several minutes [32], and, for the analysis, a minimal recording time of a few seconds was sufficient, as seen in the current study. Among recent gait-analysis methods, our system stands out for its superior availability and convenience in clinical applications [34]. Hence, we aimed to establish a connection between the application of this measurement tool [32] and its meaningful clinical applications. The initial trial involved extracting two-dimensional characteristics of pathological gait [33], while the second trial delved into the use of AI for distinguishing various gait patterns [34]. This manuscript represents our third publication, focusing on sensing the pathology of the temporospatial aspect of patients’ gait.

The role of the trunk is recognized as one of the key components of the gait system, with many studies having been conducted on this aspect [35,36,37,38,39,40,41]. Trunk lean [35], posture [36,37], balance [38,39,40], and muscle power [41] are all associated with maintaining a healthy walking mechanism. Disturbed trunk function results in slowness of gait [37] and falls [38,41]. Additionally, the movement of the upper limbs in patients with various diseases has been suggested to be linked to gait disturbance [42,43,44,45,46]. Usually, independent analyses―for example, examinations of writing or hand-to-mouth movements―were conducted, and the connection between the upper and lower limbs was deliberated [42,43,44]. Some studies using motion capture evaluated the ranges of motion of hands, elbows, or arms during walking; however, these could not include the temporospatial concepts of the movements of each part [45,46]. Hence, for gait analysis, we aim for an inclusive examination of body parts, including the trunk and upper limbs, while also being able to observe the time course during walking.

The gaits of patients with PD have been the most common ones subjected to study, mainly using the methods mentioned above to evaluate the fluctuation in motion [22,24,26,28,31]. In addition to reduced stride length and gait variability, PD patients showed impaired fractal scaling of the gait [24,26,28], which means that the normal healthy noise (fluctuation) captured during gait cycles was decreased in PD patients. The same tendency was also seen in aged people [29], patients with multiple sclerosis [23], and those with multiple-system atrophy [28]. Although the mechanism for changing this fluctuation during gait was not unclear, this approach has an impact on the research on gait dynamics, considering the underlying neural systems [19,22,25,31] and the development of rehabilitation strategies [26]. We used this method because it has the potential to assess gait dynamics using data from our system, including up to 27 body positions. This approach enabled us to evaluate the upper body’s movements, a previously unexplored area during walking.

On the other hand, the gaits of iNPH patients have not been assessed based on their fluctuation in dynamics. The pathological gait specific to iNPH is characterized by freezing gait, wide-based gait, short steps (or senile gait with reduced stride length), shuffling gait (diminished step height), instability (unsteady gait), gait festination, difficulty changing direction, and difficulties in standing up [1,8,47,48]. Video-recorded gait performance before and after the spinal tap test and shunt surgery has been recommended for gait assessments in patients with iNPH [8,47]. We have prioritized the examination of the iNPH gait [33], primarily due to the fact that the features and diagnosis of gait in iNPH remain unclear, often being referred to as just “gait disturbance”. This time, we could focus on the gaits of iNPH patients to find their characteristics and the difference compared to PD patients, and on utilizing a convenient system that requires only a brief recording of a few seconds.

The study aimed to identify temporospatial abnormalities during walking-in patients with iNPH and PD, as well as to discern the differences between the two diseases. Additionally, the study sought to explore the measurement and analysis of whole-body movement using a noninvasive iPhone application, making it an easily accessible tool for all.

## 2. Materials and Methods

### 2.1. Ethical Approvals

The study design and protocol of this study were approved by the ethics committee for human research at Nagoya City University Graduate School of Medical Science (IRB number: 60-22-0111), Shiga University of Medical Science (R2019-337), and Yamagata University School of Medicine (protocol code: 2021-10). All volunteers and patients participated in this study after providing written informed consent. The study design was prospective and observational. This study was conducted according to the approved guidelines of the Declaration of Helsinki.

### 2.2. Study Population

We collected information on age, disease history, and gait from May 2021 to November 2022 at Yamagata University Hospital and Takahata Town Hospital. The subjects with PD consisted of 23 patients with clinically established PD diagnosed according to the MDS clinical criteria for Parkinson’s disease [49]. Patients with iNPH consisted of 23 probable or definite iNPH diagnosed according to the Japanese guidelines and the management of iNPH (3rd edition) [9]. The controls were 92 healthy volunteers who participated in local health checkups and who did not have a neurodegenerative disease. Gait trials of patients were examined several times on different days. The gait of a volunteer was recorded at a specific time. To be included in either group, an individual was required to be capable of walking independently and safely for several minutes; using a single-point cane was the only assistance allowed. This criterion leads to the selection of patients with relatively mild motor symptoms.

One iNPH and two PD patients had mild resting and postural hand tremors at a rate of 1 on the Unified Parkinson’s Disease Rating Scale (UPDRS). No patients expressing dyskinesia during walking were included. Since the study focused solely on the gait capabilities during the recording time, potential influencing factors on mobility, such as medications, the timing of medication intake, shunt treatment, or other examinations, were not controlled for in this study.

### 2.3. Data Acquisition of Estimated Three-Dimensional Relative Coordinates during 1 m Circle Walking

To fit the frame of the application, which was placed about 3 m away from the gait trail, the gait of each participant was recorded. They walked in a 1 m circle for 1–3 laps, clockwise and counterclockwise (Figure 1 shows a sample of the gait circle; patients were recorded in the daily consultation room).

The TDPT-GT application could measure the 3D relative coordinates of the human body at 30 frames per second (fps) with 448 × 448 pixels and RGB color using an iPhone camera without any markers for motion capture. Details of the technology of this application and data acquisition were described in our prior publication [32]. The TDPT-GT application estimates the 3D relative coordinates of the following 24 body points: the nose, navel, and bilateral points, such as the eyes, ears, shoulders, elbows, wrists, thumbs, middle fingers, hips, knees, heels, and toes—calculated coordinates of 3 body points—the center of the head, neck, and buttocks. The middle fingers and the thumbs were captured around the metacarpophalangeal joint. We extracted the sequential 128 frames (approximately 4 s) at every point during walking in a 1 m circle, when the data is as stable as possible, except at the beginning and the end of walking. In the study, body positions were classified into the trunk positions (Figure 2): the head, neck, nose, navel, eyes, ears, shoulders, hips, buttocks, and knees, and the limb positions: elbows, wrists, thumbs, middle fingers, heels, and toes (Figure 2). The positions of the upper body were marked above the navel, while the positions of the lower body were below the navel (Figure 2).

### 2.4. Fluctuation in Body Positions during Walking

From 128 frames of gait cycles, body-position coordinates were calculated as the distances of 26 body positions from the navel, where the Fourier transform done with Python 3.10.0 produced body-position data regarding the length from the navel to the other 26 points. A slope (α) was obtained from the approximate line of the graph plotted by coordinates with the log of the power spectral density against the log of frequency (Figure 3).
$$\begin{array}{c} P(f)=kf^{\alpha }\\ P:power;\;f:\;frequency\\ logP(f)=\alpha logf+logk\\ \alpha =fluctuation\;index,in\;this\;study \end{array}$$

In this study, the slope (α) of each set of body-position data was defined as a “fluctuation index” based on the previous reports where they mentioned gait fractality [13,14,15]. We employed the value of the slope (α) to represent the fluctuation. This process was performed by clinically blinded engineers (S.S. and T.F.).

### 2.5. Statistical Analysis

For the averages of the slopes in every body position, the differences between PD, iNPH, and controls were tested by one-way ANOVA, of which the Tukey test was used to compare each of the two groups. Statistical analysis was performed by Rcmr version 2.8-0 on R version 4.2.2, and the significant level was defined as 5%.

## 3. Results

### 3.1. Clinical Characteristics

The total gait trials were 117, 56, and 184 for 23, 23, and 92 patients with iNPH, PD, and the controls, respectively (Table 1). Their average ages and standard deviations were 77.0 ± 6.4, 70.1 ± 6.0, and 72.3 ± 6.3, respectively, and they were not age-matched against both patient groups. Symptoms of iNPH were clinically known to fluctuate with the time and day; therefore, we collected several trials from an individual, each with a different date.

### 3.2. Fluctuation Index for Each Body Position

In the average of the slope (α), as the fluctuation index for each body position, significant differences were found between iNPH and the controls (*p* < 1 × 10^−7^) in all positions and those between PD and the controls (at most *p* < 1 × 10^−3^) in all positions. All absolute indices in the patient groups were smaller than those of the controls (Table 2). In the comparison between iNPH and PD, significant differences were found in seven upper body positions and eight lower body positions (*p* < 0.05) (Table 2 and Figure 4), and the differences were spreading to the right-hand area (the wrist, the middle finger, and the thumb) (Figure 4).

The absolute values in both the upper and lower limbs were smaller than those in the trunk (Figure 5). The absolute values were consistently smaller during circle walking for individuals with iNPH, PD, and the control group in all body positions. Significantly smaller absolute values for iNPH than those for PD were found in seven positions in the upper body and eight positions in the lower body (Figure 5).

## 4. Discussion

In this study, significant random fluctuations during walking were found in all body positions of iNPH and PD patients compared to the controls. Pathological fluctuation patterns, in other words, random fluctuation patterns, were observed even in the limb positions of the upper body. We could measure the pathology of the temporospatial aspect of patients’ gaits with a handy motion-capture application on a single iPhone.

### 4.1. Analysis of Gait by the System of TDPT-GT

For gait analysis, various sensors, such as pedobarography, motion capture, floor sensors, and wearable sensors, have been utilized. Optical motion-capture systems typically require the attachment of several markers to the body and the use of multiple cameras for data collection [1,2,3,4,5,6,7,8]. Time-consuming preparation, recording, and the need for large laboratory space limit their use in clinical settings and hospitals. Although wearable devices [7,16,17,18] are commonly used, they are often incapable of capturing information from multiple body parts simultaneously. Clinical demands have highlighted the necessity of a sensor that is noninvasive, requires short recording times, occupies minimal space for examination, and provides comprehensive systemic information about the body. TDPT-GT, a markerless motion-capture system, meets these requirements by enabling gait recording by capturing videos on an iPhone and instantly generates coordinates for 27 body points.

Utilizing TDPT-GT, this study aimed to identify temporospatial abnormalities in the gaits of patients with neurodegenerative diseases. In the realm of time-series analysis, detrended fluctuation analysis (DFA) is a statistical method that evaluates the self-affinity of a signal and was originally rooted in physics. Its application extends across diverse research domains, including gene analysis [50]. To quantify how the dynamics fluctuate over time while walking, DFA was used for gait analysis in the 1990s [51] and also heart-rate analysis [52]. These fluctuations are described as being (1) uncorrelated white noise, (2) long-range correlations with a power-law scaling called pink noise [51] (or 1/f noise), or (3) Brownian noise [52], corresponding to an intentional random walk (generated in a healthy person or composed of artificially shuffled data). In a normal gait pattern, complex fluctuations of an unknown origin appear as (2), which corresponds to a 1/f-like noise [51]. Older people or individuals with disease showed (1) white noise; in other words, their variation of cycles was too random [51,52]. In the present study, we did not apply the exact DFA, which needed an integrated time series split into equal boxes, where a least squares line was used to fit the data [51,52]. The gait analysis in the present study was also referenced from the DFA. Because our gait data initially exhibited significant variability due to the circular walking of many body positions, we processed a whole of 128 frames (less than 5 s) and gathered information from multiple body positions without overlays in each position. Our approach to data capture and analysis was efficient for comparing the gaits of patients and controls.

Our recordings were characterized by individuals walking in circles. This walk was necessary within the hospital’s consultation room to facilitate a seamless examination for patients with limited mobility. It was also essential to ensure stable recording within one camera frame. This was atypical in earlier gait research, which primarily relied on an analysis conducted in large laboratory rooms. That made it difficult to compare results with those of the previous gait analysis, which typically involved walking in straight lanes. However, circle walking posed a considerable challenge for patients from a medical perspective. This type of walking includes turning movements that require additional balance tasks beyond simple walking. As a result, circle walking likely induced gait disorders more sensitively than straight walking, making it a potentially valuable assessment tool. Until now, circular walking in conjunction with TDPT-GT has been effective in recording pathological gait, as demonstrated in previous studies [32,33,34]. The present study further supports this.

We were able to detect the dysfunction in the movement of the trunk and upper body including the hands, during walking with this system of TDPT-GT. The mechanism of trunk control and movement of the upper limbs in older individuals or those with various diseases has been suggested to be linked to gait disturbance [37,38,41,42,43,44,45,46]. The present markerless motion-capture technology enabled the simultaneous measurement of trunk, upper limb, and lower limb movements and allowed for a comprehensive analysis over time.

### 4.2. Fluctuation during Walking in Patients with iNPH and PD

Our study found that the fluctuation system during walking was more impaired in iNPH patients than in PD patients. The difference in gait between iNPH and PD patients has been discussed in some prior research [1,2]; however, no previous reports assessed fluctuation. Both diseases shared reduced gait velocity, due to a diminished and highly variable stride length [1]. Specific features of iNPH were broad-based with outwardly rotated feet, diminished stride length [1,2], gait velocity, and disturbed equilibrium [2]. A study assessed a series of iNPH patients and found that about 30% of complications of PD (synucleinopathy) [42]. Some patients possibly have comorbid iNPH and PD, which are impossible to differentiate; our study did not include comorbid cases. Regarding gait dynamics, we found that the fluctuated body movement of iNPH patients tended to be more impaired than that in the PD patients, meaning that this iPhone application could possibly be used as a tool for differential diagnosis.

What made the difference in fluctuation during walking between iNPH and PD patients? The external cues only mildly improved the gait disturbance in iNPH patients, whereas they were highly effective in raising the stride length and cadence in PD patients [2]. In PD patients, walking with fixed-tempo rhythmic auditory stimulation can improve many aspects of gait timing; however, it lowers fractal scaling (away from a healthy 1/f structure) [53], which is an unhealthy pattern. The new, interactive rhythmic auditory stimulation could re-establish healthy fluctuational gait dynamics in PD patients [22]. The prevalence of dementia in iNPH patients is higher than that in PD patients; therefore, iNPH patients may be distracted from following the instructions of auditory stimulation. Brain networks are supposed to generate, control, and adjust the fractal function [54] captured by signals such as imaging [55], electroencephalography [56], or connectivity [57] as well as the function expressed in gait [21,30,51]. In other words, the healthy brain has fluctuations of 1/f. In contrast to these various biological functions of the brain [54,55,56,57], dementia diminishes the brain construction associated with the fractal formation of neurons and networks [21,30,54,58]. From the present study, it can be seen that a greater impairment of the entire brain may lead to more pronounced impaired fluctuations in gait, highlighting distinctions between iNPH and PD.

We were surprised to observe random fluctuations in the positions of the upper body, including the hands, among the patient groups during walking. The significance of the right hand in the difference between iNPH and PD patients was probably due to the recording condition during the circular walking without controlling the direction of the rounds; therefore, if we controlled that, the difference might disappear between the left and right hands. Although it was possible that the potential differences in laterality within each disease group could have affected the results, we were unable to conduct a subanalysis due to the lack of precise data on laterality. When evaluating the gait of patients, the upper body or the arms are not often the subject of extensive focus. In particular, the characteristics of gait disturbance in iNPH are known to affect their lower body movements. The characteristics of gait disturbance in iNPH are especially known to affect their lower body movements [8]. A few studies have evaluated the upper limb function with a peg board in iNPH patients [59,60], where it was improved along with gait after shunt surgery [60]. However, the association between dexterity in the upper limbs and the fluctuation of motor control during walking is still unclear. Our previous study, using the same motion-capture system, used AI to differentiate between pathological gait and normal gait [34], and AI emphasized utilizing the trunk positions of the body [34]. Thus, we speculated that the trunk-posture adaptation [61] and/or the subjective vertical position [62] play a role in the movement of total or other body positions and contribute to gait or balance (Figure 6). Kinetic problems, including problems with muscle tone, induce posture impairment [63]; this may be associated with the muscle tonus symptoms of PD and iNPH, such as akinesia, dyskinesia, or paratonia (Figure 6). Although the fractality or fluctuation pattern of the upper limb has not been explored previously, using this system, we were able to detect it among patients with PD and iNPH from temporospatial perspectives.

### 4.3. Prospects for Using the Technology

The future development of TDPT-GT may include the additional automatic function associated with this fluctuation index. Moreover, incorporating the ability to sense correlations between upper and lower limb indices will contribute to advancements. Optional applications working on TDPT-GT, designed to produce seamless recording and display of indices, empower the users to provide feedback on gait information anytime and anywhere. It is likely to be beneficial for determining overall body balance during walking for all patients and older people.

The current application has the potential to be employed in diagnosing or assessing the outcomes after therapy or rehabilitation. This is crucial for preventing falls in patients with iNPH [7,8,10], PD [3,4,64,65], other diseases, or old age [66]. Fallers tended to present high consistency with power spectral density in the mediolateral axis [67]. The unpredictable body movements experienced during walking by patients can lead to imbalance and potential falls. However, patients may not always express or report these symptoms consistently. Hence, integrating the TDPT-GT application to identify bodily fluctuations could serve as a helpful tool for detecting challenging symptoms. Moreover, these less apparent symptoms require thorough assessment before and after rehabilitation. The existing system can be applied effectively in these specific areas.

### 4.4. Limitations

The limitations of the study stem from the use of the motion-capture system, which generated estimated 3D-relative coordinates. Previously, we demonstrated the correlations of each coordinate against VICON (Oxford, UK) as part of establishing full reliability and validity [32]. However, for the current study, it was necessary to analyze each coordinate of each body position independently, specifically focusing on temporospatial changes. This approach was efficient in assessing the observed differences between the disease groups and the control group. The consistent distinct trends between the disease groups and controls might suggest that the coordinates were suitable for this type of analysis.

## 5. Conclusions

By employing TDPT-GT, a user-friendly motion-capture system application on the iPhone, we succeeded in sensing the disrupted fluctuations in the movement of the entire upper and lower body, the trunk, and limbs, during walking in patients with iNPH and PD. The present study may provide new insights into gait analysis, including whole-body movements and their dynamics for patients.

## Figures and Tables

**Figure 1 sensors-23-09263-f001:**
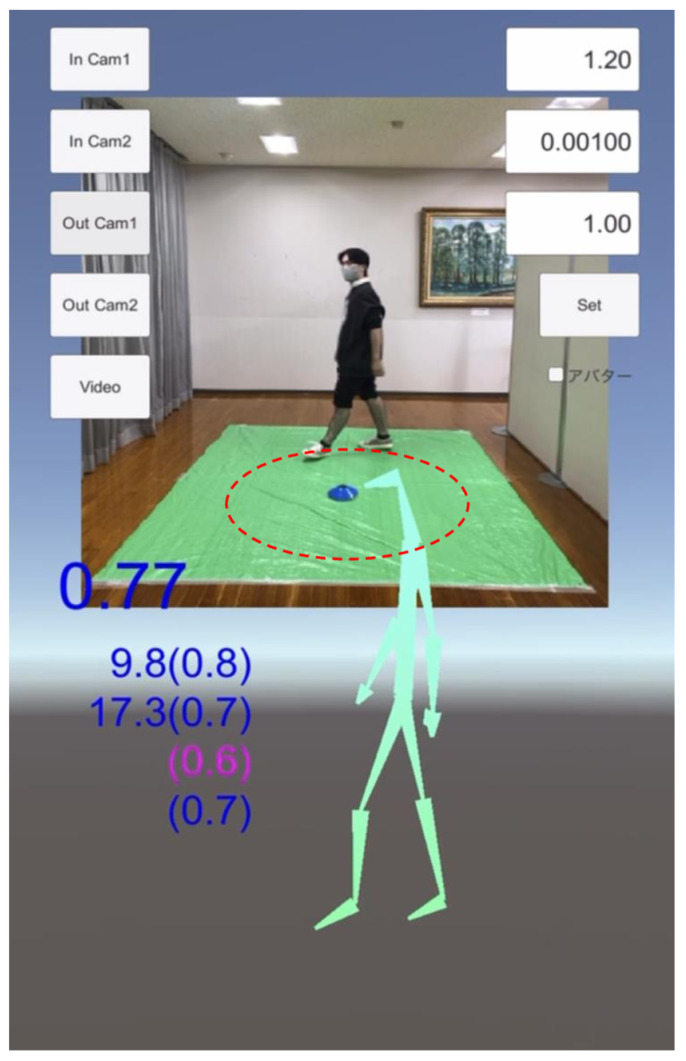
Sample picture of the gait recording on the iPhone application, TDPT-GT (Three-Dimensional Pose Tracker for Gait Test). Participants were recorded with a smartphone while walking in a 1 m circle (line of red dots) at a slow and comfortable speed. The stick figure is constructed with the 24 body points calculated by the deep leaning system of this application. The non-English word at the right side is the option button for changing the skeleton figure to a doll-like animation figure.

**Figure 2 sensors-23-09263-f002:**
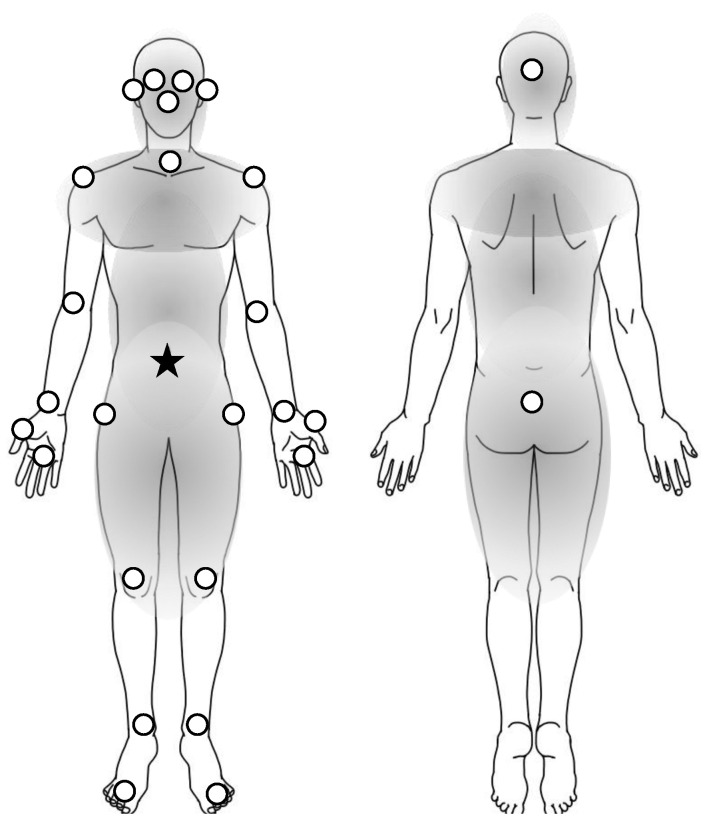
Twenty-six body positions analyzed in the study. White dots are the positions in the study. In the definition of the study, the trunk positions are included in the parts of gray color, and the limb positions are the others on the extremities. The distance of each position was calculated by the coordinates from the navel (star) to each position (dots). The positions of the upper body indicated the dots above the navel (star) and those of the lower body were below the navel.

**Figure 3 sensors-23-09263-f003:**
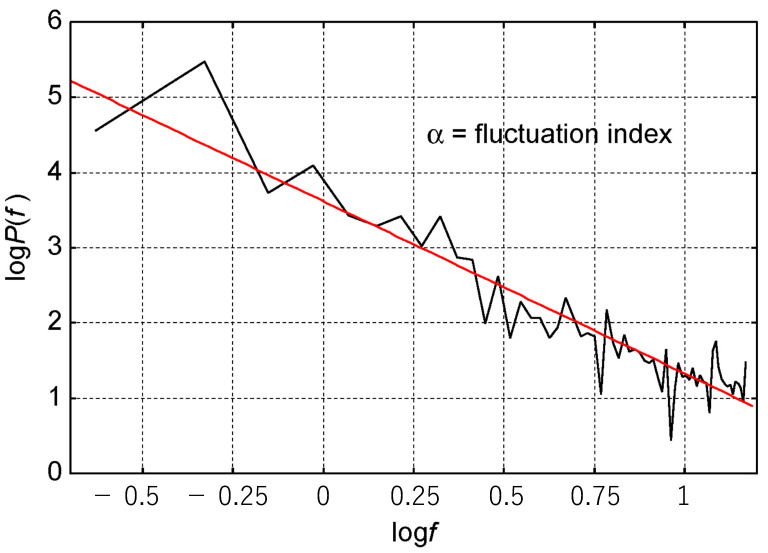
The slope as a fluctuation index. For each set of body-position data, the Fourier transform was completed, and the log of power and the log of frequency correlation were drawn (the black line). The slope (the red line) of the approximate line was a fluctuation index in the study.

**Figure 4 sensors-23-09263-f004:**
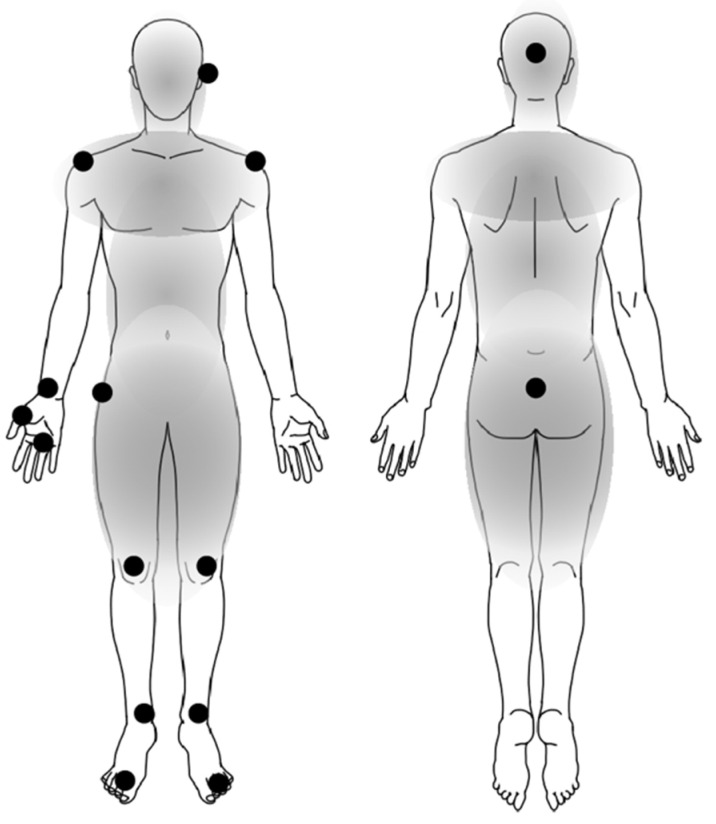
The body positions of impaired fluctuation in patients with iNPH compared with PD. The black circles (●) represent the body positions with significantly smaller absolute fluctuation indices in iNPH compared with PD. The fluctuation values are shown in Table 2. These body positions with different indices between iNPH and PD were spreading to the lower and upper body, including the right-hand area.

**Figure 5 sensors-23-09263-f005:**
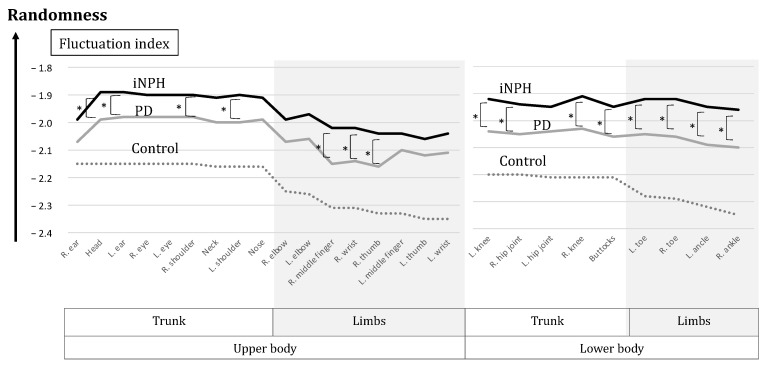
The graph of fluctuation indices. The black line indicates iNPH, the gray line indicates PD and the dotted line indicates the controls. The absolute values were consistently smaller for individuals with iNPH, PD, and the control in this order, across all body positions. The absolute values of iNPH were always smaller than those of PD, with the significances of 7 positions in the upper body and 8 positions in the lower body (*).

**Figure 6 sensors-23-09263-f006:**
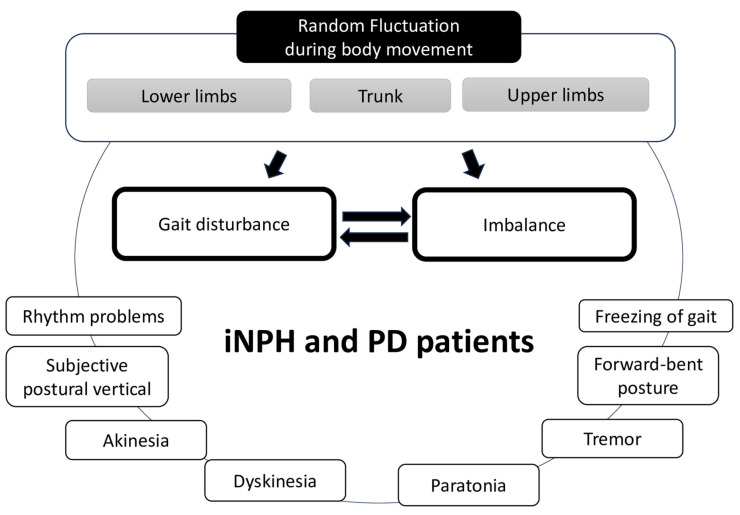
Schema of the suspected development of gait disturbance and imbalance affected by the random fluctuation of systemic body parts during gait in patients with iNPH and PD. Other symptoms of the diseases affected each other.

**Table 1 sensors-23-09263-t001:** Clinical characteristics of three datasets.

	iNPH	PD	Control
Number	23	23	92
Sex (male/female)	16/7	13/10	36/56
Number of gait trials	117	56	184
Age (average ± SD)	77.0 ± 6.4	70.1 ± 6.0	72.3 ± 6.3

iNPH, idiopathic normal pressure hydrocephalus; PD, Parkinson’s disease; SD, standard deviation.

**Table 2 sensors-23-09263-t002:** The averages of fluctuation indices in the three groups of controls, PD, and iNPH, and the *p* value from the comparison of PD and iNPH.

			Controls	PD	iNPH	*p*
Upper body	Trunk	R. ear	−2.15	−2.07	−1.99	0.101
		Head	−2.15	−1.99	−1.89	0.036 *
		L. ear	−2.15	−1.98	−1.89	0.044 *
		R. eye	−2.15	−1.98	−1.90	0.094
		L. eye	−2.15	−1.98	−1.90	0.089
		R. shoulder	−2.15	−1.98	−1.90	0.044 *
		Neck	−2.16	−2.00	−1.91	0.059
		L. shoulder	−2.16	−2.00	−1.90	0.024 *
		Nose	−2.16	−1.99	−1.91	0.118
	Limbs	R. elbow	−2.25	−2.07	−1.99	0.196
		L. elbow	−2.26	−2.06	−1.97	0.102
		R. middle finger	−2.31	−2.15	−2.02	0.035 *
		R. wrist	−2.31	−2.14	−2.02	0.049 *
		R. thumb	−2.33	−2.16	−2.04	0.045 *
		L. middle finger	−2.33	−2.10	−2.04	0.408
		L. thumb	−2.35	−2.12	−2.06	0.494
		L. wrist	−2.35	−2.11	−2.04	0.272
Lower body	Trunk	L. knee	−2.20	−2.04	−1.92	0.023 *
		R. hip joint	−2.20	−2.05	−1.94	0.019 *
		L. hip joint	−2.21	−2.04	−1.95	0.093
		R. knee	−2.21	−2.03	−1.91	0.015 *
		Buttocks	−2.21	−2.06	−1.95	0.019 *
	Limbs	L. toe	−2.28	−2.05	−1.92	0.038 *
		R. toe	−2.29	−2.06	−1.92	0.012 *
		L. ankle	−2.32	−2.09	−1.95	0.018 *
		R. ankle	−2.35	−2.10	−1.96	0.015 *

The order of lows was the ascending averages of absolute fluctuation indices in the controls. The significant differences between iNPH and the controls were with *p* < 1 × 10^−7^ in all positions, and those between PD and the controls were at least with *p* < 1 × 10^−3^ in all positions. In the comparison between iNPH and PD, significant differences were found in 7 upper body positions and 8 lower body positions (* *p* < 0.05). iNPH, idiopathic normal pressure hydrocephalus; PD, Parkinson’s disease.

## Data Availability

Data are contained within the article.

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
