# Peer review of "Fluctuations in Upper and Lower Body Movement during Walking in Normal Pressure Hydrocephalus and Parkinson’s Disease Assessed by Motion Capture with a Smartphone Application, TDPT-GT"

_sensors, 2023, doi:10.3390/s23229263_

Round 1
Reviewer 1 Report
Comments and Suggestions for Authors
Whereas this is certainly an interesting article, this reviewer has a few questions. Please provide the anti-parkinsonian medication the PD patients were on. As PD patients may have motor fluctuations, please provide the time of the assessment with respect to the last anti-parkinsonian drug intake and clarify if they were in the on- or off-state. How were tremor, rigidity and dyskinesia (if applicable) were factored in the analysis? Could sub-analyses be conducted so that the healthy controls can be age- and sex-matched? What medication were the iNPH patients on, i.e. any dopamine-related medication? Do any of the control subjects have a history of neuropsychiatric disease? This reviewer would like to disagree that clinicians do not focus on the arm swing in the upper body when assessing gait patterns, as arm swing may enable one to differentiate between diseases? Of note, PD often starts on one side of the body; could that be a reason why there was a discrepancy between PD and iNPH patients? Were the analyses performed in a blinded fashion?
Comments on the Quality of English LanguageThe wording and some typos should be corrected.
Author Response
To reviewer 1Whereas this is certainly an interesting article, this reviewer has a few questions.
→ From authors:
We appreciate your comment. We have made the best revision thanks to your pointing out.
Reviewer’s comment: Please provide the anti-parkinsonian medication the PD patients were on. As PD patients may have motor fluctuations, please provide the time of the assessment with respect to the last anti-parkinsonian drug intake and clarify if they were in the on- or off-state.
What medication were the iNPH patients on, i.e. any dopamine-related medication?
→ From authors: Thank you very much for the important notice on the clinical side. If the time after the medication and the stage of the Parkinson's disease were controlled, their gait was still diverse as a nature of this disease. We wanted to know and analyze diverse and general styles of symptoms in gait, therefore we did not emphasize that. We added the mention in the manuscript in the method section of
2.2. with red letters, line 157, page 4. Thank you.
How were tremor, rigidity and dyskinesia (if applicable) were factored in the analysis?
→ From authors: In the chapter of 2.2, we added the sentence, line 157-, page 4, which thanks to your notice. About the tremor, we wanted to sub-analyze but the patients who has this symptom could not be gathered in the study, therefore it is one of the future themes. Rigidity is primarily the most subjective and equivocal symptom; therefore, we did not employ for the basis of sub analysis.
Reviewer’s comment: Could sub-analyses be conducted so that the healthy controls can be age- and sex-matched?
→ From authors: I appreciate your attention to detail. In this study, there was some variation in the age of the patient groups. Consequently, creating age-matched controls for each disease group would have resulted in two distinct control groups, which might seem unusual. Given that we gather relatively older individuals for both the control and disease groups, and considering that individuals with the disease exhibited gait disturbances regardless of their age and sex, we decided not to match controls based on age and sex.
Do any of the control subjects have a history of neuropsychiatric disease?
→ From authors: It was written in a sentence in 2.2. study population (line 149, page 4). The participants with some neuropsychiatric diseases were not included.
This reviewer would like to disagree that clinicians do not focus on the arm swing in the upper body when assessing gait patterns, as arm swing may enable one to differentiate between diseases?
Of note, PD often starts on one side of the body; could that be a reason why there was a discrepancy between PD and iNPH patients?
→ From authors: We appreciate the professional note. We added the mention in the 5th paragraph of the discussion in line 358-, page 12.
Were the analyses performed in a blinded fashion?
→ From authors: It was naturally blinded because the data generated was just digits of all 3D coordinates. And the clinical-blinded engineers analyzed the data. We added the referral in the method section in line 211, page 6; thank you.
Once again, thank you for your valuable contribution and support to our manuscript.
Reviewer 2 Report
Comments and Suggestions for Authors
The article is devoted to analyzing gait of patients with Parkinson's disease and normal pressure hydrocephalitus. The measures were done using an iPhone with TDPT-GT application, responsible for capturing body motion and determining the coordinates of body points.
While this is a well-conducted research, my main concern is how much it fits the journal scope: the discussion and conclusion sections as they are written now make it look medical research on Parkinson's disease and normal pressure hydrocephalitus (which, of course, used some sensors). I don't see any conclusion and much discussion about sensors; the authors are concentrated on diseases. The same is true for introduction, which concentrates more on methods of analyzing gait than sensors. The authors conclude that "The present study may provide new insights into the gait dynamics impaired in patients." - which is a medical result, not sensors result. The sensor-related conclusion "These data could be quantitatively measured using a smartphone. " is too weak for an article. The authors rely on their previous article https://www.mdpi.com/1424-8220/22/14/5282 describing their application that was a good fit for Sensors, but the new study has entirely different goals.
I'd suggests the authors to either choose a medical-oriented journal from MDPI list or rewrite their article so that it will be a study of sensors, not diseases. That can be done by analyzing the study results in a different way, asking different research questions and providing different discussion. What can you conclude about sensor capabilities of your software-hardware system in that task? Can you compare it to other sensors either used to analyze patients gait or simply other motion-capturing systems in the context of diagnosing iNPH and PD? These would be sensor-related research goals. Specifically I propose:
1. Rewriting introduction with clearly formulated research questions related to sensors.
2. Rewriting the Discussion section to discuss the used sensor hardware-software system, compare it with other sensors, what can be learned about their applicability for medicine and so on.
3. Rewriting conclusions so that chief conclusions will be about sensors.
4. Describing potential usage of your findings in diagnosing and/or rehabilitation.
There are smaller concerns about this study:
1. The authors don't provide any information on the accuracy of the AI-based application they used and do not discuss its possible errors as threats to validity of their research. They cite their prior work about the used application which states "The reliability and validity of the 3D relative coordinates estimated by the TDPT-GT application have not yet been fully verified in this study." That clearly is not enough to take the AI-led "measurement" results for granted. I suggest either clearly citing prior research showing the accuracy of the used application or thoroughly discuss the threats to validity of this research because of the application inaccuracy, and how reliable the conclusions are in the light of those threats.
2. The authors captured their patients moving in circle, but I see no extensive discussion on why moving in circle was selected (e.g., the advantages over moving in a straight line, etc.) and will the results be different if the other kind of walking was used. Also, how the changing of body position during walking in circles affected the data? For example, the authors repeatedly mention the right-hand position, but during walking in circle, for some time the right hand can be hidden from the camera by the patient's body. How that affects the measurement results?
This is an important article that I want to see published, but published in the right venue according to the study goals. That will let it reach the wider audience.
Comments on the Quality of English LanguageThe article is generally understandable, but now and then it contains sentences that are significantly wrong in English grammar. For example,
1. "How to (??) describe and measure the gait of patients has been a clinical challenge. " (did you mean "The way of describing and measuring ... has been a ... challenge"?).
2. "By applying the finding fractality (??), healthy and pathological gaits" ("Applying the finding"?? what does that mean? "found fractality"?)
3. "the gait dynamics impaired in patients." (are patients impaired, or is the dynamics impaired in them?)
This list is not exhaustive. I suggest using a proofreading service or asking a colleague with native English to check your manuscript.
Author Response
To reviewer 2
Reviewer’ s comment: The article is devoted to analyzing gait of patients with Parkinson's disease and normal pressure hydrocephalitus. The measures were done using an iPhone with TDPT-GT application, responsible for capturing body motion and determining the coordinates of body points. While this is a well-conducted research, my main concern is how much it fits the journal scope: the discussion and conclusion sections as they are written now make it look medical research on Parkinson's disease and normal pressure hydrocephalitus (which, of course, used some sensors). I don't see any conclusion and much discussion about sensors; the authors are concentrated on diseases. The same is true for introduction, which concentrates more on methods of analyzing gait than sensors. The authors conclude that "The present study may provide new insights into the gait dynamics impaired in patients." - which is a medical result, not sensors result. The sensor-related conclusion "These data could be quantitatively measured using a smartphone. " is too weak for an article. The authors rely on their previous article https://www.mdpi.com/1424-8220/22/14/5282 describing their application that was a good fit for Sensors, but the new study has entirely different goals.
I'd suggests the authors to either choose a medical-oriented journal from MDPI list or rewrite their article so that it will be a study of sensors, not diseases. That can be done by analyzing the study results in a different way, asking different research questions and providing different discussion. What can you conclude about sensor capabilities of your software-hardware system in that task? Can you compare it to other sensors either used to analyze patients gait or simply other motion-capturing systems in the context of diagnosing iNPH and PD? These would be sensor-related research goals. Specifically I propose:
- Rewriting introduction with clearly formulated research questions related to sensors.
→From authors to reviewer:
We appreciate your valuable suggestions. Our team, comprising engineers, application developers, data analysts, and clinical medical doctors, focuses on gait analysis. Our project primarily revolves around addressing clinical inquiries and needs, particularly for patients experiencing mobility challenges, such as those with walking difficulties or a high risk of falls due to various diseases. Our initial focus is on how to accurately detect and cater to their needs. Subsequently, we delve into discussions about the sensing and measurement aspects.
Since 2017, our team has consistently followed this trajectory, resulting in the publication of three articles in your journal, utilizing the motion capture system known as TDPT-GT, which is also used in the current study. The start was from the article of the basic establishment of detailed this motion capture system (Aoyagi et al. doi:10.3390/s22145282.) Then, we engaged to link this new sensing tool to clinical meaningful use as convenient as possible. The first link succeeded in finding the distinctive characteristics of pathological gait observed through this system (Yamada et al. doi:10.3390/s23020617). The second link was able to connect to patients by making a new AI distinguishment system of gait patterns using the data collected by this system (Iseki et al. doi:10.3390/s23136217). This manuscript marks our third publication on the combination of sensing the pathology of the temporospatial aspect of movement and the pathology of gait. Thanks to your suggestions, this flow was mentioned in introduction and discussion, line 87- page 2, and line 278-, page 11 of revised manuscript.
- Rewriting the Discussion section to discuss the used sensor hardware-software system, compare it with other sensors, what can be learned about their applicability for medicine, and so on.
→From authors to reviewer:
We find it challenging to effectively articulate the connection between the sensor and its clinical application. While your advice (reviewer 2) primarily focuses on the sensing aspect, the reviewer 1 suggested emphasizing the commonalities within a clinical trial. The clinical information and discussion might seem extensive for you (reviewer 2), but these aspects were emphasized as crucial requirements by reviewer 1. However, we appreciate you two reviewers for giving us the chance to make a comprehensive article for readers of both sides, sensing and clinical meaning or developing. We revised the article considering that, such as in the discussion part divided in subchapters of 4.1. Analysis of gait and the system of TDPT-GT (line 276-, page 11) and 4.2. Fluctuation during walking in patients with iNPH and PD (line 323, page 12). The subchapter of 4.1 discussed associated with sensing and that of 4.2 treated the clinical aspects.
3Rewriting conclusions so that chief conclusions will be about sensors.
→From authors to reviewer:
In line 405-, page 15, we revised the conclusion in the basis of newly framed discussions of 4.1 and 4.2.
4 Describing potential usage of your findings in diagnosing and/or rehabilitation.
→From authors to reviewer:
Appreciating your suggestion, we made the new subchapter in the discussion as 4.3. Prospects for using the technology (line 382-, page 14) to discuss diagnosing and rehabilitation.
There are smaller concerns about this study:
- The authors don't provide any information on the accuracy of the AI-based application they used and do not discuss its possible errors as threats to validity of their research. They cite their prior work about the used application which states "The reliability and validity of the 3D relative coordinates estimated by the TDPT-GT application have not yet been fully verified in this study." That clearly is not enough to take the AI-led "measurement" results for granted. I suggest either clearly citing prior research showing the accuracy of the used application or thoroughly discuss the threats to validity of this research because of the application inaccuracy, and how reliable the conclusions are in the light of those threats.
→From the authors to the reviewer: We appreciate your processional note. We added the mention as this discussion as 4.4 limitations in line 393-, page 14.
- The authors captured their patients moving in circle, but I see no extensive discussion on why moving in circle was selected (e.g., the advantages over moving in a straight line, etc.) and will the results be different if the other kind of walking was used. Also, how the changing of body position during walking in circles affected the data? For example, the authors repeatedly mention the right-hand position, but during walking in circle, for some time the right hand can be hidden from the camera by the patient's body. How that affects the measurement results?
→From the authors to the reviewer:
In the previous manuscript, we addressed the issue of circular walking in the limitations section of the discussion, similar to your concern. Your observation prompted us to elevate this discussion to a more prominent position in the overall discussion (line 310-, page 11).
This is an important article that I want to see published, but published in the right venue according to the study goals. That will let it reach the wider audience.
→From the authors to the reviewer:
We appreciate your highly professional comments. We have made the best revision thanks to your pointing out.
Comments on the Quality of English Language
The article is generally understandable, but now and then it contains sentences that are significantly wrong in English grammar. For example,
- "How to (??) describe and measure the gait of patients has been a clinical challenge. " (did you mean "The way of describing and measuring ... has been a ... challenge"?).
→ From the authors to the reviewer:
We appreciate your suggestion for formal English. We revised that part.
- "By applying the finding fractality (??), healthy and pathological gaits" ("Applying the finding"?? what does that mean? "found fractality"?)
→ From the authors to the reviewer:
We rewrote as “By applying the fractality.”
- "the gait dynamics impaired in patients." (are patients impaired, or is the dynamics impaired in them?)
→ From the authors to the reviewer:
We rewrote as “the gait dynamics impairment in patients”.
This list is not exhaustive. I suggest using a proofreading service or asking a colleague with native English to check your manuscript.
→ From the authors to the reviewer: Thank you for the suggestion. Surely, the previous manuscript was after the native check. And this revision was also screened by native editors again. I hope the quality of English improved.
Once again, thank you for your valuable contribution and support to our manuscript.
Round 2
Reviewer 2 Report
Comments and Suggestions for Authors
The main problem of this article in my opinion is still its scope and the scope of the journal. It is hard to see the contribution of this article to researching sensors (while its contribution to researching diseases and gait is unquestionable). I appreciate being informed about your research plans, and your previous article suited Sensors well because they concentrated on the technical aspects of the sensors and applications you used, but this article has a diferrent goal.
One of the established research practices is formulating a clear set of Research Question (RQ) in Introduction and then discuss the study results regarding them in Discussion section.
So I recommend adding one or more sensor-related research questions to the Introduction section and then discussing the answers in the Discussion section to emphasize the new contribution to the research field of sensors.
Comments on the Quality of English LanguageEnglish (especially in the new text) still has significant problems. For example, "Our research appears to suggest(??) that ", usage of the verb "sense" in "we succeeded in sensing the disrupted fluctuations in the movement", "use of the motion capture system, which generated estimated 3D relative coordinates" (generated estimated coordinates??) and so on.
Author Response
Reviewer 2
The main problem of this article in my opinion is still its scope and the scope of the journal. It is hard to see the contribution of this article to researching sensors (while its contribution to researching diseases and gait is unquestionable). I appreciate being informed about your research plans, and your previous article suited Sensors well because they concentrated on the technical aspects of the sensors and applications you used, but this article has a diferrent goal.
One of the established research practices is formulating a clear set of Research Question (RQ) in Introduction and then discuss the study results regarding them in Discussion section.
So I recommend adding one or more sensor-related research questions to the Introduction section and then discussing the answers in the Discussion section to emphasize the new contribution to the research field of sensors.
→From the authors to the reviewer
We appreciate your comment and advice very much. However, our results were shown the same way; we tried to summarize our scope and goal along with the scope of “Sensors.” In parts of the abstract (page 1), the introduction (page 3, line 106-), the discussion (page 12, line 340-, and page 14, line 409-), and the conclusion (page 15, line 441-), we added the scope and the goal (highlighted with a yellow marker). Thanks to your advice, we supposed that our goals could be shown clearly.
From the reviewer:
English (especially in the new text) still has significant problems. For example, "Our research appears to suggest(??) that ", usage of the verb "sense" in "we succeeded in sensing the disrupted fluctuations in the movement", "use of the motion capture system, which generated estimated 3D relative coordinates" (generated estimated coordinates??) and so on.
→From the authors to the reviewer
We appreciate your note and advice. The manuscript was edited in the third round. The final editing was by MDPI official editing service. I hope the language will satisfy you and the readers.
